# Effect of Feeding Improved Grass Hays and *Eragrostis tef* Straw Silage on Milk Yield, Nitrogen Utilization, and Methane Emission of Lactating Fogera Dairy Cows in Ethiopia

**DOI:** 10.3390/ani10061021

**Published:** 2020-06-11

**Authors:** Shigdaf Mekuriaw, Atsushi Tsunekawa, Toshiyoshi Ichinohe, Firew Tegegne, Nigussie Haregeweyn, Nobuyuki Kobayashi, Asaminew Tassew, Yeshambel Mekuriaw, Misganaw Walie, Mitsuru Tsubo, Toshiya Okuro, Derege Tsegaye Meshesha, Mulugeta Meseret, Laiju Sam, Veerle Fievez

**Affiliations:** 1United Graduate School of Agricultural Sciences (UGSAS), 4-101 Koyama-Minami Tottori-shi, Tottori University, Tottori 680-8553, Japan; 2Amhara Region Agricultural Research Institute, Andassa Livestock Research Center, P.O. Box 27, Bahir Dar, Ethiopia; misganaw2000@gmail.com (M.W.); mulugeta12andassa@gmail.com (M.M.); 3Arid Land Research Center, Tottori University, 1390 Hamasaka, Tottori 680-0001, Japan; tsunekawa@tottori-u.ac.jp (A.T.); kobayashi.nobuyuki@alrc.tottori-u.ac.jp (N.K.); tsubo@tottori-u.ac.jp (M.T.); 4Faculty of Life and Environmental Science, Shimane University, Matsue, Shimane 690-8504, Japan; toshi@life.shimane-u.ac.jp; 5College of Agriculture and Environmental Sciences, Bahir Dar University, P.O. Box 5501, Bahir Dar, Ethiopia; firewtegegne@yahoo.co.uk (F.T.); yeshambel166@gmail.com (Y.M.); asaminew2@gmail.com (A.T.); deremesh@yahoo.com (D.T.M.); laijusam@yahoo.com (L.S.); 6International Platform for Dry Land Research and Education, Tottori University, 1390 Hamasaka, Tottori 680-0001, Japan; nigussie_haregeweyn@yahoo.com; 7Laboratory of Landscape Ecology and Planning, Graduate School of Agricultural and Life Sciences, The University of Tokyo, Tokyo 113-8657, Japan; aokuro@mail.ecc.u-tokyo.ac.jp; 8Department of Animal Sciences and Aquatic Ecology, Ghent University, 9000 Gent, Belgium; Veerle.Fievez@ugent.be

**Keywords:** dairy production, dairy feeding, dryland, improved forage, methane emission, nutrient excretion, precision feeding, plasma metabolite, smart farming, total mixed ration

## Abstract

**Simple Summary:**

In tropical regions, it is common practice to feed dairy cows poor-quality roughage, but this diet has been shown to decrease animal productivity and increase methane emissions. For these reasons, introducing alternative roughage feeds, such as nutritionally improved forages or bio-chemically treated straw, is essential for improving milk yield, dietary nitrogen utilization, and reducing enteric methane emission from dairy cows. Thus, we evaluated the effects of natural pasture hay, two improved grass hays (Napier and *Brachiaria* hybrid grasses), and treated teff straw silage feeding as basal diets on nutrient digestibility, milk yield, nitrogen utilization efficiency, and enteric methane emissions using lactating Fogera dairy cows. Our results showed that improved grass hays and treated teff straw silage diet feeding increased milk yield, nutrient digestibility, and nitrogen utilization efficiency as compared to natural pasture hay. Moreover, the cows fed with improved grass hays and treated teff straw resulted in changing the nitrogen excretion pathway from urine to feces, as well as reduction of the methane production per daily milk yield. Hence, these results provide a novel feeding regimen through feeding nutritionally upgraded forages as a basal diet, which improves milk yield, nutrient utilization efficiency, and reduction of methane emission for sustainable dairy production in tropical regions.

**Abstract:**

The nutritionally imbalanced poor-quality diet feeding is the major constraint of dairy production in tropical regions. Hence, alternative high-quality roughage-based diets are required to improve milk yield and reduce methane emission (CH4). Thus, we tested the effects of feeding natural pasture hay, improved forage grass hays (Napier and *Brachiaria* Hybrid), and treated crop residues (*Eragrostis tef* straw) on nutrient digestibility, milk yield, nitrogen balance, and methane emission. The eight lactating Fogera cows selected for the experiment were assigned randomly to a 4 × 4 Latin square design. Cows were housed in well-ventilated individual pens and fed a total mixed ration (TMR) comprising 70% roughage and 30% concentrate. The four roughage-based basal dietary treatments supplemented with formulated concentrate were: Control (natural pasture hay (NPH)); treated teff straw silage (TTS); Napier grass hay (NGH); and *Brachiaria* hybrid grass hay (BhH). Compared with the control diet, the daily milk yield increased (*p* < 0.01) by 31.9%, 52.9%, and 71.6% with TTS, NGH, and BhH diets, respectively. Cows fed BhH had the highest dry matter intake (8.84 kg/d), followed by NGH (8.10 kg/d) and TTS (7.71 kg/d); all of these intakes were greater (*p* = 0.01) than that of NPH (6.21 kg/d). Nitrogen digestibility increased (*p < 0.01*) from the NPH diet to TTS (by 27.7%), NGH (21.7%), and BhH (39.5%). The concentration of ruminal ammonia nitrogen was higher for cows fed NGH than other diets (*p* = 0.01) and positively correlated with plasma urea nitrogen concentration (R² = 0.45). Feeding TTS, NGH, and BhH hay as a basal diet changed the nitrogen excretion pathway from urine to feces, which can help protect against environmental pollution. Estimated methane yields per dry matter intake and milk yield were decreased in dairy cows fed BhH, NGH, and TTS diets when compared to cows fed an NPH diet (*p* < 0.05). In conclusion, feeding of TTS, NGH, and BhH roughages as a basal diet to lactating dairy cows in tropical regions improved nutrient intake and digestibility, milk yield, nitrogen utilization efficiency, and reduced enteric methane emission.

## 1. Introduction

Limited availability of high-quality feed is the major stumbling block affecting dairy cattle production in tropical regions [1]. Moreover, owing to an alarmingly rising human population, grazing lands used for natural hay production for the dry season are widely converted to croplands and plantations [2], thereby forcing cattle to graze on marginal and overgrazed lands with poor-quality forage [3,4]. This practice is further aggravated by the fact that the yield and nutritive value of tropical grasses decline sharply as the dry season approaches, leading to reduced feed intake and milk production in conventional grazing systems [5,6]. In addition, compared with those elsewhere, dairy animals in developing countries produce more methane and excrete more nitrogen per unit of animal product, primarily because of the feeding of low-quality roughage, which is unbalanced in nutrients [7]. This situation is most severe in countries like Ethiopia, which has a huge livestock population [8], and farmers typically feed dairy cows arbitrarily, without balancing the diet, thus decreasing potential milk production and increasing methane emission [9]. Hence, proposing an appropriate feeding regimen that enables the reduction of methane emissions and mitigates environmental N pollution from dairy cattle production remain major concerns [6,10,11].

One potential strategy for better feeding of dairy cows during periods of critical feed shortage, particularly in the dry season, is to use improved forage varieties and enhance the quality of available crop residues [12]. To this end, supporting the traditional feed resources with planted forages appears to be a plausible and sustainable solution [11]. This situation encourages the cultivation of suitable and adapted improved forages to the tropical regions, such as Napier grass (*Pennisetum purpureum*) [13,14] and *Brachiaria* hybrid (CIAT 36087) Mulato II grasses [15], to fulfill the nutrient requirements of dairy cows for milk production [16]. Napier grass is a fast-growing perennial grass widely grown for smallholder dairy production in tropical and subtropical regions [14,17], while *Brachiaria* hybrid grasses are the newest options for improving productivity in semi-intensive systems, and some of these cultivars are high-yielding, nutritious, and ecofriendly [18]. As a result of these benefits, *Brachiaria* hybrid grasses have recently garnered considerable interest in Africa and have factored into several ongoing initiatives to support the emerging livestock industry in tropical regions, particularly to prepare hay for the dry season [19,20]. However, despite the potential of these improved forages in various tropical regions, including Ethiopia [21], information regarding their contribution to the performance of lactating dairy cows and methane emission is sparse. Moreover, the previous common feeding practice of dairy production in tropical regions is separate forage and concentrate feeding but, in this study, a total mixed ration (TMR) was used to allow for the incorporation of the basal and supplemented feed ingredients.

As another option, crop residues inherently poor in nutritive values represent a large roughage feed resource for ruminant animals, especially during the dry season in tropical regions [22]. However, most smallholder farmers currently feed untreated crop residues, consequently diminishing dairy cows’ production performance [23]. Thus, treating these feed resources using bio-chemicals, such as effective microbes (i.e., a liquid mixture of important beneficial microorganisms) and urea molasses, improved feed digestibility and nutritive value of straw [24]. Therefore, in the current study, we assessed the effect of treating teff straw with effective microbes, molasses, and urea on the enhancement of the nutrient utilization of the poor-quality roughage feed available, as well as the effect on the performance of lactating dairy cows [25,26]. 

We hypothesized that using improved forage (grasses, hay, and treated teff straw silage) would increase the performance of dairy cows and reduce methane emission as compared with natural pasture hay. Therefore, the objective of this study was to evaluate the effects of feeding improved grasses, hay, and treated teff straw silage as a basal diet on milk yield, nutrient intake and digestibility, dietary nitrogen utilization efficiency, and methane emission. Hence, considering the similar situation in Ethiopia in which the feed priority for livestock is focused on introducing and promoting improved forages to enhance dairy production [12], this study proposes an alternative, improved forage hay-based feeding regimen for critical feed-shortage periods in the dry season. The findings also contribute to informing policy makers and development practitioners regarding future feeding strategies for dairy production.

## 2. Materials and Methods 

### 2.1. Experimental Location, Cows, and Design 

The study was conducted at the Andassa Livestock Research Center of the Amhara Region Agricultural Research Institute, which is 1730 m above sea level in the Bahir Dar Zuria (11°42′–11°92′ N, 37°07′–37°65′ E) district of the Amhara region. It receives an average annual rainfall of 1150 mm, and temperatures range from 6.5 to 30 °C [27].

We obtained eight Fogera dairy cows in their second lactation stage with 120 ± 18 days in milk, an average body weight of 238.3 kg, and 2.4 parity from Andassa Livestock Research Center dairy farm. The lactating dairy cows were placed in individual separate pens in a well-ventilated barn with a concrete floor and an appropriate drainage slope. All cows were weighed and drenched with broad-spectrum anthelmintic before the start of the experiment. The experimental design was a replicated 4 × 4 Latin square design [28]. The experiment consisted of 4 treatments and 4 periods; the treatments were assigned randomly for dairy cows within each period. Each of the 4 periods had 2l days in duration, comprising 14 days of TMR dietary feed adaptation and 7 days of data collection. The experiment was conducted for a total of 12 weeks. All animal care and handling procedures were reviewed, and the experimental protocols were approved by the Andassa Livestock Research Center prior to conducting the experiment, and the animals were under constant observation of veterinarians. 

### 2.2. Experimental Dietary Treatments and Feed Management

The four roughage-based basal dietary treatments supplemented with formulated concentrate in TMR bases were: Control (natural pasture hay (NPH)); treated teff straw silage (TTS); Napier grass hay (NGH); and *Brachiaria* hybrid grass hay (BhH). The natural pasture hay was purchased from a private dairy farm and was composed predominantly of grass species (*Andropogon*, *Cynodon*, *Digitaria*, *Hyparrhenia*, and *Panicum* spp.) and legumes (*Trifolium quartinianum*, *T. polystachyum*, and *Indigofera atriceps*) as characterized by Denekew et al. [29]. The improved forages were planted in accordance with the recommended agronomic practices for Napier grass (accession number 1574) [13,30] and *Brachiaria* grass (accession number CIAT 36087) [15] in 2.65 ha of irrigated land at the Andassa Livestock Research Center. The forages were irrigated to support continuous growth for harvesting sufficient amounts of hay. The harvested forage was air-dried for 2–3 days in the field and mixed before storage for the feeding trial. The hay was chopped into 2–5 cm lengths and mixed with the concentrate in the TMRs. For chemical analysis, samples of each forage were taken and oven-dried to a constant dry weight, ground to pass through a 2-mm screen, and then stored in airtight plastic bags. For teff straw treated silage preparation, 1 L of effective microbes, 1 kg of molasses, 2.5 kg of urea, and 18 L of chlorine-free water were mixed and used to treat 50 kg of teff straw as described previously by Dejene et al. [31]. Treated teff straw silage was put into airtight plastic bags and well packed to avoid trapping air. To facilitate anaerobic fermentation, filled bags were kept at room temperature (25 °C) for 21 days, as recommended for environmental conditions such as those in an Ethiopian climate [26]. The formulated concentrate mixture was purchased from a private animal-feed factory company and consisted of maize (*Zea mays*) (40%), noug (*Guizotia abyssinica*) seed cake (49%), wheat bran (*Triticum Aestivum*) (8%), iodized salt (1%), and ruminant premix (2%). The ruminant premix (produced by INTRACO Ltd., Antwerp, Belgium) contained the following additives (per kg): Ca, 1310.5 g; Na, 192.2 g; Mg, 520.8 g; Fe, 5000 mg; Mn, 5000 mg; Zn, 10,000 mg; I, 150 mg; Se, 40 mg; Co, 15 mg; vitamin A, 999,750 IU; vitamin D3, 199,950 IU; vitamin E, 800 mg; butylated hydroxytoluene, 50 mg; and ethoxyquin, 55 mg. 

Cows were fed TMR diets consisting of 70% roughage and 30% concentrate on a dry matter (DM) basis. To achieve the targeted milk yield (4 kg milk/d), the TMR diets were formulated based on National Research Council [32] for the maintenance and lactation requirement of the lactating dairy cow (average body weight, 238.3 kg) for each treatment group. The TMR diets were offered in the morning (8:00 h), at noon (12:00 h), and during the afternoon (16:00 h). Cows had individual and free access to drinking water throughout the entire experiment. The chemical composition of feed ingredients and the TMR diets are presented in Table 1. The acid detergent lignin (ADL)/neutral detergent fiber (NDF) ratio of the dietary treatments for NPH, TTS, NGH and BhH is 0.13, 0.17, 0.22, and 0.25, respectively.

### 2.3. Measurements and Sample Collection 

The TMR feed offered and the orts were recorded daily at each feeding time over the whole experimental period. From day 15 to 21 of each experimental period, samples of the TMR offered and ort were collected and stored for laboratory analysis. Daily milk yields for all 8 cows were recorded at each milking time. Cows were hand-milked twice daily, and the milking time was in the morning (08:00 h) and afternoon (16:00 h) throughout the experiment. During each period (from 15 to 21 days), milk samples were collected at each milking time (08:00 h and 16:00 h), pooled by cow, and then transported in an ice box for milk composition analysis. Another pooled milk sample for each period was stored immediately at −20 °C for determination of milk urea nitrogen (MUN) content. 

Spot urine samples from all cows were collected in plastic containers between days 15 and 21 of each experimental period three times per day (07:00, 13:00, and 17:00 h) and immediately stored at –20 °C for determination of N and creatinine contents. A sample of 100 mL urine was collected from every cow and 8 mL of an aqueous solution of sulfuric acid 10% (v/v) added, in which outflowing air was led to trap aerial ammonia and reduce urine pH to below 3, as described previously [33]. Fecal samples were collected on days 15 through 21 of each experimental period by direct sampling from the rectum of each cow in the morning (07:00), afternoon (13:00), and evening (17:00 h). Composite fecal samples from each cow in each period were maintained at −20 °C until laboratory analysis. 

After feeding the cows on the twenty-first day of each experimental period, blood samples were collected from the jugular veins of individual animals into 10 mL sodium heparin and potassium EDTA vacuum tubes, as described previously by Nichols et al. [28]. The blood samples were pooled over sampling time points according to cow and period; plasma was prepared by centrifuging blood at 1000× *g* for 5 min at 23 °C; the supernatant was transferred to identified plastic tubes and stored at –20 °C until laboratory analysis. Rumen fluid samples were collected from each cow two hours after the morning feeding on the twenty-first day of each experimental period by inserting a rumen-fluid collector through an esophageal gavage with a manual sucker [34]. The samples were temporarily placed on ice and then processed for ruminal ammonia nitrogen analysis. The body weight (BW) of each dairy cow was weighed at the start, middle, and end of each period for the whole experiment time by using a ground weight balance; the weight of cows was obtained by averaging the weights taken before feeding and after milking on two successive days. 

### 2.4. Laboratory Analyses and Procedures

All feed and fecal samples were oven-dried at 60 °C for 48 h. The dried samples were ground to pass through a 2-mm screen and stored in plastic bags for subsequent determination of chemical components. The DM and organic matter (OM) contents of the diets and feces were determined according to AOAC [35]. The crude protein (CP) concentration was estimated by multiplying the N concentrations by 6.25, and N concentrations were determined according to the Kjeldahl method [36]. The neutral detergent fiber (NDF), acid detergent fiber (ADF), and acid detergent lignin (ADL) contents of the feed and feces were determined according to the procedures of Goering et al. [37]. The metabolizable energy (ME) concentration (MJ/kg DM) was estimated on the basis of 24-h gas production (mL) from in vitro gas fermentation [38];
ME = 2.20 + 0.136 × GP + 0.057 × CP + 0.0029 × CP^2^(1)
where GP is the 24-h gas production volume (mL/0.2 g DM) and CP is the crude protein content (%) of the feed. Estimated net energy maintenance (NEm) and net energy lactation (NE_l_) were calculated according to the following equation [32]: NEm = ME (0.554 + 0.287 × ME/GE)(2)
NE_l_ = ME × (0.4632 + 0.24 × ME/GE)(3)
where the gross energy (GE) concentrations of feed were determined by using a bomb calorimeter (CA-4AJ, Shimadzu Corporation, Kyoto, Japan). For the TMR, these parameters were calculated according to the ratio composition of roughage and concentrate ingredients. Analysis of in vitro gas production for the feed samples was conducted at Shimane University (Japan) in accordance with the procedure described by Mekuriaw et al. [39]. 

Feed intake was determined on days 16 through 21 of each period as the difference between the weight of the TMR feed offered to the animals and that left unconsumed: Daily nutrient intake (kg/d) = ([DM offered (kg/d) × % of nutrient in TMR] – [DM remaining (kg/d) × % of nutrient in orts])(4)
whereas nutrient digestibility was calculated as ([nutrient intake (kg/d) – fecal nutrient output (kg/d)]/nutrient intake) × 100% [40]. 

Milk fat, protein, and lactose contents were determined by using a LactoScan milk analyzer (Milkotronic Ltd., Nova Zagora, Bulgaria). Fat- and protein-corrected mik production (FPCM) (kg/d) was calculated according to De Koster et al. [41]:FPCM = (0.337 + 0.116 × milk fat [%] + 0.06 × milk protein [%]) × kg of milk yield(5)

The milk yield efficiency was calculated according to Nichols et al. [10]: Efficiency = milk yield (kg)/DMI (kg)(6)

The milk urea nitrogen (MUN) concentrations were obtained through spectrophotometric enzymatic colorimetric methodology using commercially available kits (urease–GIDH method, *λ* = 525 nm; product code 410-55391 _ 418-55191, Fujifilm, Wako, Japan). 

Urinary creatinine concentrations (mg/dL) were determined by using commercial enzymatic colorimetric assay kits (sarcosine oxidase method, *λ* = 515 nm; product code 439-90901, Fujifilm, Wako, Japan) and UV–VIS spectrophotometry (DR6000, Hach, Dusseldorf, Germany). Total daily urine volume was estimated by dividing daily urinary creatinine excretions by the observed creatinine concentration of spot urine samples, assuming a daily creatinine excretion of 0.197 ± 0.047 mmol/kg BW [42]. Urinary N was analyzed according to the Kjeldahl method [35] by using the same equipment for feed, feces, and milk samples; hence, urinary N excretion was calculated by multiplying urinary N by urine volume. Nitrogen excreted in milk was calculated by using the equation [43]:Milk N (g/d) = milk CP concentration (g/kg) × milk yield (kg/d)/6.38(7)

The N excreted in feces was determined as [44]: Fecal N (g/d) = CP in feces (g/kg) × DM fecal excretion (kg/d)/6.25(8)

Fecal output was estimated by using chromium oxide as an external indicator according to Kimura et al. [44]. Chromium oxide was given daily (5 g for each cow) in the morning feeding on days 15 through 21 of each experimental period. The N balance was obtained by subtracting the values for N in urine, feces, and milk from the total N intake in grams [43]:N retention = N intake − N fecal − N urine − N milk(9)

Plasma underwent determination of glucose, non-esterified fatty acid (NEFA), β-hydroxybutyrate (BHBAA), and blood urea nitrogen (BUN) contents. These parameters were measured by using commercial enzymatic colorimetric assays (glucose: mutarotase-GOD method, product code 439-90901, *λ* = 455 nm; NEFA: ACS-ACOD method, product code 279-75401, *λ* = 550 nm; BHBAA, cyclic enzyme method, product code 279–75401, *λ* = 405 nm; and BUN: urease-GIDH method, product code 410–55391 _ 418–55191, *λ* = 340 nm; all from Fujifilm) and a UV–VIS spectrophotometer (DR6000, Hach). Immediately after collection, rumen fluid was tested for pH by using a handheld portable pH meter and then stored at −20 °C until analysis of ruminal ammonia nitrogen concentration. For ruminal fluid samples that were preserved with 1% H_2_SO_4_, the rumen fluid collected was centrifuged at 2000× *g* for 15 min and the resulting supernatant was analyzed for ruminal ammonia nitrogen concentration, as described previously [45].

### 2.5. Estimation of Enteric Methane Emission 

The enteric methane emissions of lactating dairy cows were calculated according to a recommended intercontinental equation Mutian et al. [46]: CH_4_ production (g/day per cow) = 124 (±10.44) + 13.3 (±0.32) × DM intake (DMI, kg/day)(10)

This estimation equation was chosen because of its low root mean square prediction error (RMSPE = 17.5) compared with those of other equations in the same publication. Correspondingly, Appuhamy et al. [47] also evaluated the performance of more than 40 empirical models in predicting enteric CH_4_ emissions and suggested that using DM intake alone can be sufficient for satisfactory enteric methane emission prediction.

### 2.6. Statistical Analysis

Data regarding feed intake, nutrient digestibility, nitrogen balance, plasma metabolites, milk yield, and milk composition were analyzed through analysis of variance (ANOVA) using the mixed model procedure of SAS (version 9.4; SAS Institute Inc., Cary, NC, USA). The model is:Y*_ijk_* = µ + T*_i_* + P*_j_* + C*_k_* + ε*_ijk_*,(11)
where, Y*_ijk_* represents the observation on cow *k* given treatment *i* at period *j*;

µ is the overall mean;

T*_i_* represents the fixed effect of the *i*th diet treatment, *i* = 1 to 4; 

P*_j_* represents the random effect of the *j*th period, *j* = 1 to 4; 

C*_k_* represents the random effect of the *k*th cow, *k* = 1 to 2; 

ε*_ijk_* is the random residual error.

The model contained treatment as fixed effects, period and cow as random effects. Differences among the means were considered significant at the *p* ≤ 0.05 level according to Tukey’s test. To assess the relationships among plasma urea concentration, milk urea concentration, and ruminal ammonia N, as well among methane emission and milk yield, regressions were fitted by using the pooled data of all dietary treatments.

## 3. Results

### 3.1. Feed Intake and Nutrient Digestibility

The nutrient intake of the lactating Fogera dairy cows fed the different roughage-based basal total mixed ration diets, and their digestibility, are presented in Table 2. Compared with those given the control diet (NPH), the cows that consumed TTS, NGH, and BhH diets had higher (*p =* 0.01) dry matter intake (DMI), by 1.50, 1.89, and 2.63 kg/d, respectively. Cows fed with BhH had the highest (*p* < 0.05) DM, OM, and N intake, followed by NGH and TTS. Likewise, cows fed with BhH, NGH, and TTS had higher (*p* < 0.05) DM, OM, and N diet digestibility than those fed with NPH. Cows fed with BhH had increased average daily gain (ADG) (*p* = 0.02) followed by NGH and TTS, as compared to NPH. The digestibility of N increased (*p* < 0.01) by 27.79%, 21.7%, and 39.5% from NPH to the TTS, NGH, and BhH diets, respectively. On the other hand, cows fed NPH had higher NDF (*p* = 0.03) and ADF (*p* = 0.01) intake, whereas lower NDF (*p* = 0.02) and ADF (*p* = 0.01) digestibility than those fed the other treatment diets. Cows fed with BhH had the highest GE (*p* = 0.01) and ME (*p* < 0.01) intake, followed by NGH and TTS. 

### 3.2. Nitrogen Balance and Utilization Efficiency

Nitrogen intake increased by 61.8%, 57.6%, and 97.4% with the TTS, NGH, and BhH diets, respectively, compared with the NPH diet (*p* < 0.01; Table 3). Nitrogen retention showed a similar trend and was greater (*p* < 0.01) by 174.0%, 160.2%, and 264.2% for cows fed TTS, NGH, and BhH, respectively, than those fed the NPH diet. The milk nitrogen of cows fed TTS, NGH, and BhH was increased (*p =* 0.02) by 6.8, 9.9, and 15.1 g/d, respectively, over that for those fed the NPH diet. The urinary N:fecal N ratios decreased (*p =* 0.02) by 10.5%, 34.6%, and 38.2% from NPH to TTS, NGH, and BhH, respectively. Urinary N:N intake ratios showed a similar trend and were decreased (*p =* 0.04) by 29.5%, 44.0%, and 53.0% NPH for TTS, NGH, and BhH, respectively, compared with those with the NPH diet. 

### 3.3. Plasma Metabolites and Ruminal Fermentation Characteristics

Cows fed with TTS had the highest (*p* < 0.01) plasma urea nitrogen concentration among the dietary treatments (Table 4.). In contrast, the lowest (*p* = 0.01) plasma glucose content was recorded for TTS than other diets. Remarkably, cows fed NPH had higher (*p* ≤ 0.05) NEFA and BHBA than those fed other diets. Ruminal ammonia N was higher (*p* = 0.01) in cows fed NGH than in those fed the other diets. Rumen pH was not affected by dietary treatments (*p* ≥ 0.05). Ruminal ammonia N was positively correlated with plasma urea nitrogen (*R*² = 0.53; Figure 1a). 

### 3.4. Milk Yield and Composition 

Compared with cows fed NPH, daily milk yield was increased (*p* = 0.01) by 31.98%, 52.97%, and 71.60% in cows fed TTS, NGH, and BhH, respectively (Table 5). Fat- and protein-corrected milk showed a similar increasing (*p* < 0.02) trend among the dietary treatments (Table 5). Although dietary treatment had no effect on the concentrations of milk fat (*p* = 0.12) and protein (*p* = 0.06), TTS and NGH increased (*p* = 0.04) milk lactose, and MUN was greater (*p* = 0.04) in cows fed TTS compared with other diets. Cows fed NGH had a higher milk urea nitrogen concentration (MUN) compared with other diets. The MUN was positively correlated with plasma urea (*R*² = 0.45) (Figure 1b). The TTS, NGH, and BhH diets increased (*p* = 0.03) the efficiency of milk production of lactating dairy cows compared with NPH by 6.13%, 25.94%, and 23.55%, respectively. 

### 3.5. Estimated Enteric Methane Emission

Estimated methane production increased with increasing DMI by 19.95, 25.14, and 34.98 g/d/cow for TTS, NGH, and BhH, respectively, compared with NPH (*p =* 0.01; Table 6). In comparison with that for the NPH diet, CH_4_/DM intake decreased (*p =* 0.01) by 11.67%, 14.0%, and 17.85% for TTS, NGH, and BhH, respectively. Similarly, cows fed TTS, NGH, or BhH produced less CH_4_ per kg of total OM (*p =* 0.03) and GE (*p =* 0.025) intake than those fed the NPH diet. In reflecting the difference in DMI per milk yield, CH_4_ per unit of milk yield (*p =* 0.02) and FPCM (kg) (*p =* 0.02) were lower for TTS, NGH, and BhH than the NPH diet. The amount of estimated CH_4_ emitted per unit of DMI was negatively linearly correlated (*R*^2^ = 0.76) with milk yield production across the dietary treatments (Figure 2). This finding similarly showed lower estimated methane production with higher milk yield from cows fed BhH compared with other diets (*p* = 0.02). 

## 4. Discussion

### 4.1. Feed Intake and Nutrient Digestibility

Our findings support the use of improved forages (BhH and NGH) and TTS as a basal TMR diet to increase feed intake and digestibility of nutrients compared to that of NPH. Dry matter intake increased as dietary CP content increased because of the inclusion of TTS, NGH, or BhH as nitrogen sources in a basal diet. Similarly, Mutimura et al. [14] reported that the differences in DMIs of Napier grass and *Brachiari brizantha* fed as sole diets are related to the dietary composition of grasses. In this study, the improved grasses had the highest nutrient digestibility, with BhH superior to NGH. In line with this, Mutimura et al. [14] reported that, on the basis of palatability, *Brachiaria* is preferred to Napier grass. In our current study, NDF digestibility increased from NPH (48.1%) to BhH (56.5%) in TMR diets. These differences in fiber digestibility of TMR diets might be related mainly to the quality of the dietary NDF ingested [48]. Similarly, the inclusion of improved forages in the TMR diet affects DMI through their influence on nutrient digestibility [49]. Moreover, the higher intake of TTS than of NPH may reflect increased palatability due to the urea and molasses added to the diet [24]. 

### 4.2. Nitrogen Excretion and Utilization 

In this study, the dietary TMR treatments influenced nitrogen intake, excretion, balance, and utilization the efficiency for milk production (Table 3). Particularly striking is our finding that feeding TTS, NGH, or BhH hay redirected the N excretion pathway from urine to feces; a change that can benefit the environment. Urinary N is more volatile and harmful to the environment than fecal N because the urinary urea N is inorganic N; microbial ureases rapidly hydrolyze urinary urea N to ammonium, which is then converted to ammonia, leading to N loss from the farm to the environment [50]. In agreement with our results, previous studies also reported that dairy cows fed large quantities of high-quality fresh forage or hay shift their excretion of N from urine to feces and thus toward sustainable dairy production [51,52]. The urinary N to fecal N ratio decreased from NPH to the TTS, NGH, and BhH diets; this indicates that using treated teff straw silage and improved forages as a basal diet in the TMR reduces urinary N excretion and indirectly mitigates ammonia emission in dairy cows for smart farming that is environmentally-friendly. Furthermore, the N balance also varied among the dietary treatments we evaluated, in which the cows fed TTS, NGH, and BhH retained increasingly more N than those fed NPH. This variation might be due to the difference in N intake, digestibility, and excretion of the TMR diets fed for dairy cows. Therefore, the improved forage- and treated teff straw-based TMR diets we used as basal diets for dairy cows increased protein utilization efficiency and decreased the amount of N excreted to the environment per kg of milk produced. This indicates that inclusion of improved forages in the TMR diets of dairy cows can improve N efficiency [52]. 

### 4.3. Plasma Metabolites and Rumen NH_3_ Concentration 

The plasma urea nitrogen concentrations that we obtained (2.6–3.5 mmol/L) are within the acceptable values for lactating dairy cows (2.6–7.0 mmol/L) [53]. We attribute the lower plasma urea nitrogen concentrations recorded for the BhH and NGH diets to better utilization of nitrogen in the rumen; because the end-product of protein in the rumen is ammonia N, then ammonia N is absorbed by rumen epithelial cells and transferred to liver to form urea, which is released to blood in the form of blood urea [54]. This might be because the higher values of ruminal NH_3_-N in BhH and NGH diets were reflecting the higher plasma urea N in this study. The plasma glucose values in this study (2.55–3.77 mmol/L) were under the range of 2.5–4.2 mmol/L for lactation dairy cows and above the level (2.29 mmol/L) that cause hypoglycemia in early lactation cows [32]. However, the highest plasma glucose concentration recorded for the BhH TMR diet in this study might be due to a positive energy status of the cow related to a higher DMI [32]. With the exception of the higher value for cows fed the NPH diet, plasma concentrations of NEFA and BHBA in our study were within the ranges proposed for mid-lactation dairy cows [32,55]. The high NEFA concentration of the NPH diet (0.5 mmol/L) was likely due to low DMI. Previous reports of the negative correlation of DMI with NEFA and BHBA [56] support our current results. The NEFA concentration of cows fed NPH in our study exceeded the threshold (NEFA ≥ 0.5 mmol/L) that may be associated with milk loss, thus perhaps accounting for the low milk production of this treatment group [57]. Furthermore, the maximal BHBA (0.22 mmol/L) of cows fed NPH in our study was below the threshold proposed by Gruber et al. [58] (≥1.2 mmol/L). 

We found here that the increase in ruminal ammonia N concentration was directly associated with the observed plasma urea concentration (Figure 1a). In agreement with our findings, Migliano et al. [59] likewise reported a strong positive correlation between ruminal ammonia N and plasma urea N. The inclusion of improved forages in TMR diets might contribute to highly degradable protein sources that decreased ruminal ammonia N concentrations, but ruminal ammonia N likely remained adequate for microbial growth [60]. The ruminal pH that we obtained is normal for lactating dairy cows fed roughage-based diets [52], perhaps because the dietary NDF concentrations of all of the dietary treatments were sufficient to maintain the optimal ruminal pH for cows fed high-forage diets. This might be mainly due to a high forage-to-concentrate ratio (70:30) [61].

### 4.4. Milk Yield and Methane Emission 

The milk yield in the cows fed BhH was almost double that of the NPH diet. These findings indicate that improved forage inclusion in the TMR diets, such as *Brachiaria* hybrid cultivars, could increase the milk production of indigenous dairy cows. In contrast, the lowest milk yield in this study was for the cows fed the NPH diet; this might be related to the low dietary CP in this TMR, as well as the decreased DMI. Moreover, Hussien et al. [62] reported even lower milk yields (1.4 kg/d) for Fogera dairy cows fed a natural grass hay basal diet supplemented with concentrate in separate feeding. This variation might be due to the TMR feeding system used in this study that can increase the precision-feeding of dairy cows [63]. There is no TMR-based lactating dairy cow feeding system for indigenous dairy cows in tropical regions to compare the milk composition result of this study. However, it is comparable with the same dairy breed reported by Hussien et al. [62] that fed natural hay pasture and concentrate mix in separate feeding. Indicative of inefficient N utilization [64], the MUN concentration was higher for TTS than in all other diets. This might have been related to the higher amounts of CP due to the inclusion of urea in the TTS-based TMR diet that led to increased MUN excretion [32]. 

Estimated enteric CH_4_ emission is the by-product of ruminal fermentation via methanogenesis and is thus substantially affected by a range of factors, including dietary components [65]. Thus, the methane production variation in our study among the dietary treatments might be related to the difference in the proportion of nutrients, such as the high NDF content of the NPH-based TMR [11]. Specifically, the fermentation of fibrous materials favors the formation of acetate and butyrate which contribute to the production of CH_4_ [65]. In our study, methane emission per kg of OM intake was lower for the TTS, NGH, and BhH diets than for NPH, thereby indicating that highly digestible roughages are promising means for reducing CH_4_ emission. Likewise, cows fed NGH and BhH, followed by TTS, had significantly lower methane output per unit of daily milk yield than NPH diet, showing that the TMR diets containing improved forage species with high CP contents have great potential in mitigating methane emission for climate smart dairy production [11]. Moreover, the environmental cost of producing NGH and BhH forages should be examined before recommending using this strategy to mitigate CH_4_ emissions and urinary N excretion. In the present study, CH_4_ emissions from enteric fermentation were estimated using intercontinental equations because, as far as the authors’ knowledge, there is no methane emission prediction equation developed for the Ethiopian dairy breed; this may be the first to report such methane emission data. Therefore, more studies are needed to measure the actual CH_4_ emissions of the Ethiopian dairy breed, considering the negative effects of CH_4_ production on global warming. 

## 5. Conclusions

Dairy cows fed with treated teff straw (TTS), Napier grass hay (NGH) and *Brachiaria* hybrid grass hay (BhH) had better feed intake and nutrient utilization, as well as milk yields, than cows fed natural pasture hay (NPH). Feeding TTS, NGH, and BhH hay as a basal diet changed the N excretion pathway from urine to feces and indirectly mitigated ammonia emissions, thus potentially benefitting the environment. Furthermore, N utilization varied among dietary treatments; for example, the N retained in the bodies of cows fed TTS, NGH and BhH was higher than in cows fed the NPH diet. Moreover, cows fed TTS, NGH, and BhH produced significantly less estimated methane per daily milk yield and fat protein correct milk. In summary, feeding TMR diets containing improved forages and treated teff straw silage as a basal diet to dairy cows increased nutrient digestibility, milk yield, and nitrogen utilization efficiency, as well as reduced nutrient excretion and CH_4_ emission to the environment, thereby potentially improving dairy production in tropical regions. Consequently, this study highlighted the possibility of increasing production and reducing GHG emissions in countries like Ethiopia, where precision-feeding is a limitation.

## Figures and Tables

**Figure 1 animals-10-01021-f001:**
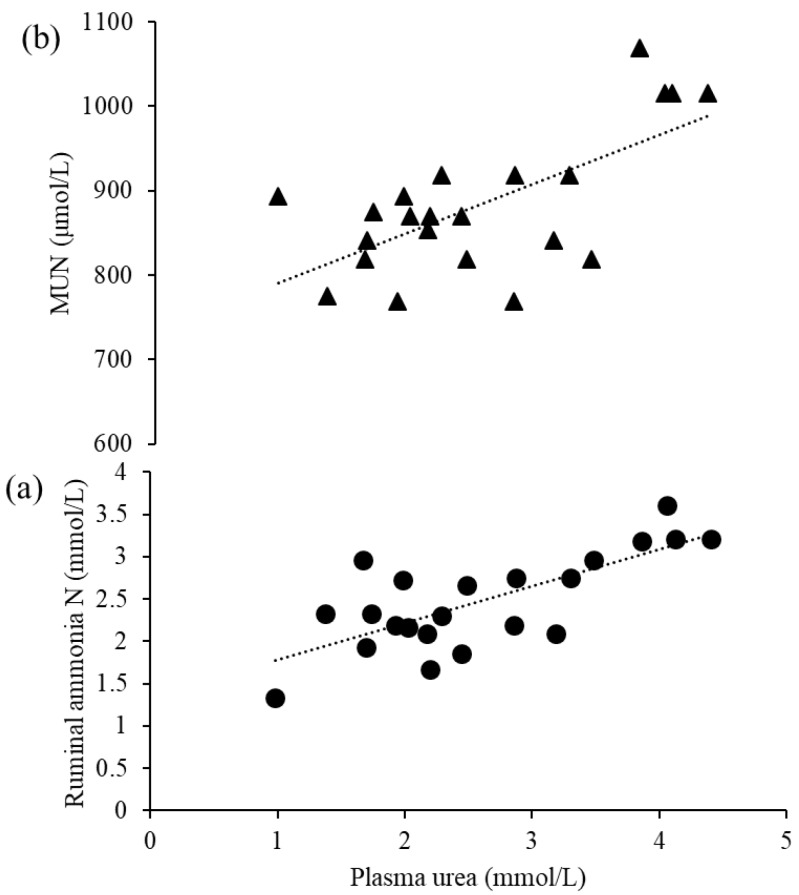
(**a**) Plasma urea nitrogen relative to ruminal ammonia N (ruminal ammonia N = 0.438 × plasma urea nitrogen + 1.3416; *R*² = 0.53); (**b**) plasma urea nitrogen relative to milk urea nitrogen (MUN = 57.507 × plasma urea nitrogen + 734.86; *R*² = 0.45).

**Figure 2 animals-10-01021-f002:**
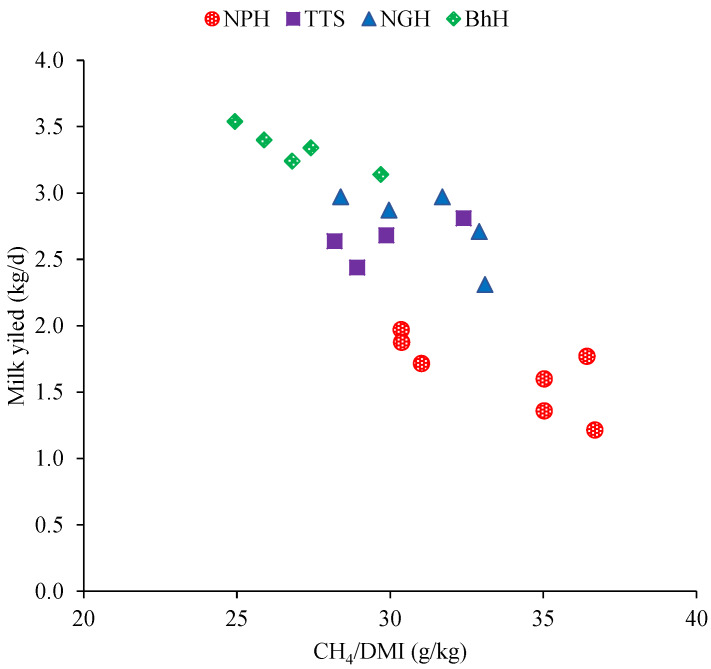
CH_4_ emissions per dry matter intake (DMI) corresponding to daily milk yield (*R*² = 0.61). NPH, natural pasture hay; TTS, treated teff straw; NGH, Napier grass hay; BhH, *Brachiaria* hybrid grass hay.

**Table 1 animals-10-01021-t001:** Chemical compositions of feed ingredients and the experimental total mixed ration diets.

Feed Ingredient	DM	OM	CP	NDF	ADF	ADL	GE	ME	NEm	NE_l_
Natural pasture hay	878.61	869.88	41.75	741.79	512.08	76.47	15.52	8.02	1.44	1.27
Treated teff straw silage	660.73	618.34	115.50	644.18	454.28	49.79	16.23	7.03	0.98	1.03
Napier grass hay	856.87	837.75	92.54	626.51	450.59	83.10	14.15	7.28	1.01	0.82
*Brachiaria* grass hay	822.71	817.89	135.90	564.16	362.58	44.56	15.88	8.01	1.33	0.99
Formulated concentrate	910.40	904.86	201.60	340.57	153.78	60.27	18.32	9.26	1.47	1.22
**Treatment Diets (g/kg DM)**										
NPH	901.21	892.46	122.90	541.21	333.03	68.43	16.94	8.51	1.45	1.25
TTS	690.13	604.44	158.14	492.46	304.08	55.04	17.36	8.14	1.23	1.13
NGH	863.65	839.38	147.82	483.63	302.29	71.71	16.28	8.27	1.24	1.02
BhH	866.58	827.91	168.34	452.47	258.20	52.23	17.01	8.62	1.40	1.11

DM, dry matter (g/kg as fed); OM, organic matter (g/kg DM); CP, crude protein (g/kg); NDF, neutral detergent fiber (g/kg DM); ADF, acid detergent fraction (g/kg DM); ADL, acid detergent lignin (g/kg DM); GE, gross energy (MJ/kg DM); ME, metabolizable energy (MJ/kg DM); NEm, net energy for maintenance (Mcal/kg DM); NE_l_, net energy for lactation (Mcal/kg DM); NPH, natural pasture hay; TTS, treated teff straw (DM/As fed); NGH, Napier grass hay; BhH, *Brachiaria* hybrid grass hay.

**Table 2 animals-10-01021-t002:** Feed intake and nutrient digestibility of lactating dairy cows fed natural pasture hay, treated teff straw, Napier grass hay, and *Brachiaria* hybrid grass hay.

Item	Dietary Treatment				SEM	*p*
	NPH	TTS	NGH	BhH		
**Intake (kg/d)**						
DM	6.21 ^c^	7.71 ^b^	8.10 ^b^	8.85 ^a^	0.56	0.01
OM	5.60 ^c^	6.26 ^b^	7.12 ^ab^	8.01 ^a^	0.46	<0.01
N	0.12 ^c^	0.20 ^b^	0.19 ^b^	0.24 ^a^	0.15	0.01
NDF	4.31 ^a^	3.87 ^b^	2.66 ^c^	2.78 ^a^	0.29	0.03
ADF	2.51 ^a^	2.63 ^a^	2.01 ^b^	1.36 ^c^	0.08	0.01
GE (MJ/d)	104.89 ^c^	132.80 ^b^	131.18 ^b^	150.58 ^a^	9.43	<0.01
ME (MJ/d)	53.38 ^b^	39.58 ^b^	41.90 ^b^	75.87 ^a^	8.30	<0.01
DMI, g/kg BW^0.75^	97.95 ^c^	115.58 ^b^	124.06 ^b^	153.97 ^a^	0.01	0.03
ADG ( g/day)	326.18 ^c^	446.38 ^b^	485.71 ^ab^	627.46 ^6a^	27.08	0.02
**Nutrient Digestibility (%)**						
DM	64.20 ^c^	79.60 ^b^	80.60 ^b^	87.70 ^a^	4.95	0.03
OM	58.90 ^c^	71.47 ^b^	74.12 ^ab^	79.39 ^a^	4.35	<0.01
N	59.70 ^c^	76.24 ^b^	72.70 ^b^	83.36 ^a^	4.94	<0.01
NDF	48.14 ^c^	52.89 ^b^	54.24 ^a^	56.65 ^a^	1.77	0.03
ADF	47.72 ^c^	49.10 ^b^	50.45 ^b^	53.42 ^a^	1.22	0.01

NPH, natural pasture hay; TTS, treated teff straw; NGH, Napier grass hay; BhH, *Brachiaria* hybrid grass hay; SEM, standard error of mean; DM, dry matter (kg/d); OM, organic matter; CP, crude protein; NDF, neutral detergent fiber; ADF, acid detergent fiber; GE, gross energy (MJ/kg DM); ME, metabolizable energy (MJ/kg DM); DMI (g/kg BW^0.75^), dry matter intake in metabolic body weight; ADG, average daily gain; DM, dry matter; N, nitrogen; ^a–c^ Means within a row with no common superscripts differ (*p* < 0.05).

**Table 3 animals-10-01021-t003:** Nitrogen utilization and balance of lactating dairy cows fed natural pasture hay, treated teff straw, Napier grass hay, and *Brachiaria* hybrid grass hay.

	Dietary Treatment				SEM	*p*
	NPH	TTS	NGH	BhH		
**Nitrogen Balance (g/d)**						
N intake, g/d	120.90 ^c^	195.59 ^b^	190.58 ^b^	238.68 ^a^	24.1	<0.01
Fecal N	48.64 ^a^	40.50 ^b^	37.10 ^c^	35.21 ^c^	4.5	0.01
Urinary N	25.70 ^b^	29.33 ^a^	22.87 ^c^	23.85 ^c^	1.9	0.04
Milk N	12.88 ^c^	19.64 ^b^	22.86 ^ab^	27.98 ^a^	2.3	0.02
N retention	33.68 ^d^	92.27 ^b^	87.64 ^c^	122.65 ^a^	17.5	0.03
**Nitrogen Utilization Efficiency (g/g)**						
Fecal N/N intake	0.40 ^a^	0.22 ^b^	0.27 ^b^	0.17 ^c^	0.02	0.03
Urinary N/N intake	0.21 ^a^	0.15 ^a^	0.12 ^b^	0.10 ^b^	0.02	0.04
Urinary N/fecal N	0.60 ^a^	0.54 ^b^	0.39 ^c^	0.37 ^c^	0.06	0.02
Milk N/N intake	0.11	0.10	0.12	0.12	0.01	0.36
N retention/N intake	0.33 ^c^	0.47 ^ab^	0.46 ^ab^	0.51	0.05	<0.01

NPH, natural pasture hay; TTS, treated teff straw; NGH, Napier grass hay; BhH, *Brachiaria* hybrid grass hay; SEM, standard error of mean; ^a–d^ means within a row with no common superscripts differ (*p* < 0.05).

**Table 4 animals-10-01021-t004:** Plasma concentrations (mmol/L) of metabolites, and ruminal fermentation characteristics in lactating dairy cows fed natural pasture hay, treated teff straw, Napier grass hay, and *Brachiaria* hybrid grass hay.

	Dietary Treatment				SEM	*p*
	NPH	TTS	NGH	BhH		
Plasma urea nitrogen	2.96 ^b^	3.36 ^a^	2.77 ^b^	2.64 ^b^	0.23	0.02
Plasma glucose	2.94 ^b^	2.55 ^c^	2.70 ^b^	3.81 ^a^	0.40	0.01
Plasma NEFA	0.51 ^a^	0.36 ^b^	0.32 ^b^	0.30 ^b^	0.39	0.02
Plasma BHBA	0.22 ^a^	0.05 ^b^	0.05 ^b^	0.04 ^b^	0.06	0.04
Ruminal ammonia N	2.29 ^c^	2.70 ^b^	3.25 ^a^	2.91 ^b^	0.21	0.01
Rumen pH	6.60	6.50	6.90	6.70	0.33	0.36

NPH, natural pasture hay; TTS, treated teff straw; NGH, Napier grass hay; BhH, *Brachiaria* hybrid grass hay; SEM, standard error of mean; NEFA, non-esterified fatty acids; BHBA, β-hydroxybutyrate; N, nitrogen; ^a–c^ Means within a row with no common superscripts differ (*p* < 0.05).

**Table 5 animals-10-01021-t005:** Milk yield, composition, and urea content of lactation dairy cows fed natural pasture hay, treated teff straw, Napier grass hay, and *Brachiaria* hybrid grass hay.

	Dietary Treatment				SEM	*p*
	NPH	TTS	NGH	BhH		
Yield						
Milk (kg/d)	1.77 ^c^	2.34 ^b^	2.71 ^b^	3.34 ^a^	0.27	0.01
FPCM (kg/d)^d^	2.06 ^c^	2.69 ^b^	2.91 ^b^	3.40 ^a^	0.38	0.02
***Composition (%)***						
Fat	5.49	6.38	5.68	5.40	0.13	0.16
Protein	2.76	2.76	2.88	2.78	0.16	0.06
Lactose	4.19 ^c^	4.81 ^a^	4.59 ^ab^	4.21 ^b^	0.23	0.04
MUN (μmol/L)	858.6 ^b^	975.6 ^a^	837.2 ^b^	879.5 ^b^	37.8	0.04
Efficiency	0.28 ^b^	0.29 ^b^	0.35 ^a^	0.34 ^a^	0.02	0.03

NPH, natural pasture hay; TTS, treated teff straw; NGH, Napier grass hay; BhH, *Brachiaria* hybrid grass hay; SEM, standard error of mean; FPCM, fat- and protein-corrected milk; MUN, milk urea nitrogen; ^a–c^ Means within a row with no common superscripts differ (*p* < 0.05).

**Table 6 animals-10-01021-t006:** Methane emission of lactating dairy cows fed natural pasture hay, treated teff straw, Napier grass hay, and *Brachiaria* hybrid grass hay.

	Dietary Treatment				SEM	*p*
	NPH	TTS	NGH	BhH		
CH_4_ (g/d)	206.5 ^d^	226.5 ^c^	231.7 ^b^	241.6 ^a^	7.66	0.011
CH_4_/BW^0.75^ (g/kg)	3.0	3.2	3.3	3.4	0.08	0.21
CH4/feed intake or milk yield (g/kg)						
CH_4_/DM intake	33.27 ^a^	29.38 ^b^	28.61 ^b^	27.33 ^b^	1.44	0.01
CH_4_/OM intake	36.87 ^a^	32.25 ^b^	32.17 ^b^	30.70 ^b^	1.28	0.03
CH_4_/ GE intake	8.21 ^a^	7.12 ^b^	7.40 ^b^	6.70 ^c^	0.33	0.03
CH_4_/milk yield	116.61 ^a^	96.89 ^b^	79.50 ^c^	71.12 ^c^	9.09	0.02
CH_4_/FPCM	84.12 ^a^	69.61 ^b^	65.20 ^b^	38.15 ^c^	8.85	0.02

NPH, natural pasture hay; TTS, treated teff straw; NGH, Napier grass hay; BhH, *Brachiaria* hybrid grass hay; SEM, standard error of mean; CH_4_, methane; BW^0.75^, metabolic body weight; DM, dry matter; OM, organic matter; GE, gross energy; FPCM, fat- and protein-corrected milk yield; ^a–d^ Means within a row with no common superscripts differ (*p* < 0.05).

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
