# Peer review of "Effect of Feeding Improved Grass Hays and Eragrostis tef Straw Silage on Milk Yield, Nitrogen Utilization, and Methane Emission of Lactating Fogera Dairy Cows in Ethiopia"

_animals, 2020, doi:10.3390/ani10061021_

Round 1

Reviewer 1 Report

Dear Authors,

I have revised the manuscript 811414 entitled : Effect of feeding improved grass hays and Eragrostis tef straw silage on milk yield, nitrogen balance, and methane emission of lactating Fogera dairy cows, Ethiopia. The manuscript is important in the field and show the extent of GHG emissions and urinary N excretion of dairy farming in Ethiopia, and how simple improvements like improving dietary forage source can have a marked effect on environmental footprint of dairy production. The study is well executed and well described. I have, however, the following concerns:

  1. More experimental details need to be added please see my specific comments.
  2. In the statistical model, Latin square design assumes no interaction between period and treatment therefore this should be removed from the model and the data should be reanalysed.
  3. Correlation analysis should be removed because of limited data points (4 cows in 4 periods), and because the correlations was established in the literature long time ago and it does not add to the scientific value of the manuscript.
  4. The high N retention values indicate either increase in body weight, or incorrect protocol to determine N intake- secretion- excretion. Please indicate in the table or in the body weights of cows (total or change) and comment on this if there was a problem in the protocol.

Specific comments

L30: introducing

L32: reduce

L40 reduced

L42 improving

L54 and through out the manuscript P values should have a consistent form with P in capital letters italic, and the value in 2 digits after the point.

L61 protection against

L66 enteric methane emission

L78: add “per unit of animal product” at the end of the line

L83 replace avoid with mitigate

L84 remove “is still”

L106 suggest rewording because it implies that you will be a comparing between treated and untreated teff straw

L111 reduce methane emission compared with…

L127 Please provide days in milk at the start of the project

L145 dried to replace wilted

L146 mixed

L154-160 On average, at 240 kg body weight, DMI should be calculated at 7.2 kg/d. Were the diet formulates so that such a level of intake would meet the nutrient requirement to produce 4 kg milk/d? Please put in details

Table 1 Identify abbreviations

Table 1 Is this on DM basis? please specify. Also report ADL on NDF basis. Also add correct units of NEm and NEL and reference of calculations

L181 replace by “treatment group” with “cow”

L184, L189, L192 what time the sampling was done? was it spread throughout the day or at specific time each day?

L256 there is no details on chromium oxide use in the previous text. Please add

L301-302: Please separate results of intake from results of digestibility. It is rather confusing in this form

L301-302 here it is CP digestibility while it is N digestibility in the table. Please correct.

L302 tended? I see it is significant please correct and add P value

L307 here and throughout the manuscript: in table title, full name of treatment should be spelled out

L319-320 in TTS, NGH and BhH, respectively compared to NPH

Table 3: what is the difference between milk N/ N intake and MNE? Please define MNE in the table footnote

Table 4: Ruminal pH and plasma concentrations…. Rumen pH values should be in 2 decimals

L347 remove that

L352 higher milk urea nitrogen

L368 previous comment regarding correlation data. GEI is similar DMI, and you are using a formula to calculate CH4 based on DMI or GEI, for sure there will be correlation previous comment regarding correlation data. GEI is similar DMI, and you are using a formula to calculate CH4 based on DMI or GEI, for sure there will be correlation

L393 avoid abbreviation at the beginning of the sentence

L397 remove potential

L400-402 hard to understand. Please separate into 2 or 3 sentences o reword accordingly.

L404-405 urea and molasses added to the tef straw silage

L408-410 also there is a positive relationship between DMI and fecal N excretion. Please use this to enrich discussion

L420-421 This is not clear, please reword and be more specific

L426 rumen NH3 concentration rather rumen fermentation characteristics

L428-431 but this is against the absence of diet effect on rumen NH3 concertation. Please elaborate on this

L435-436 This is a speculation not measured. Please specify

L440-443 add reference

L443-444 how is this related to your findings, please elaborate.

L448 plasma urea concertation

L452-455 mainly due to high forage to concentrate ratio (70:30).

L463-464 This is irrelevant. Please consider removing or rewording

L368-470 it is HIGHLY unlikely that 1 kg of molasses used in 50kg of hay to exert such an effect. Most probably it is due to a positive energy status of the cow due to higher DMI. Please re-investigate

L477-479 unclear, please reword

L48: suggest replacing over with “which contribute to”

L491 Suggest adding: The environmental cost of producing NGH, bhH forages should be examined before recommending using this strategy to mitigate CH4 emissions and urinary N excretion.

In the conclusion, please use full name of treatment rather than the abbreviation, at least when first mentioned.

In the conclusion: The key point in the conclusion that increasing production reduced enteric CH4 emission, which should be the focal point of reducing GHG emissions in countries like Ethiopia where nutrition is limitation

L494-495 This is repeated in the next phrase, please remove one of the repeated phrases or reword

Author Response

Reply to Reviewer #1 comments

  1. I have revised the manuscript 811414 entitled: Effect of feeding improved grass hays and Eragrostis tef straw silage on milk yield, nitrogen balance, and methane emission of lactating Fogera dairy cows, Ethiopia. The manuscript is important in the field and show the extent of GHG emissions and urinary N excretion of dairy farming in Ethiopia, and how simple improvements like improving dietary forage source can have a marked effect on environmental footprint of dairy production. The study is well executed and well described:

Response: We appreciate the encouraging comments on our manuscript.

I have, however, the following concerns:

  1. More experimental details need to be added please see my specific comments.

Response: We thank the reviewer for the useful comment. The necessary detail information related to the experiment was added based on the reviewer specific comments in the revised manuscript.

  1. In the statistical model, Latin square design assumes no interaction between period and treatment therefore this should be removed from the model and the data should be reanalyzed.

Response: We thank the reviewer for this suggestion. We accepted the reviewer’s comment and removed the interaction between period and treatment (Ti x Pj) from the model (see line 291-297). As well reconfirmed the analyzed data according to the revised model.

  1. Correlation analysis should be removed because of limited data points (4 cows in 4 periods), and because the correlations was established in the literature long time ago and it does not add to the scientific value of the manuscript.

Response: We appreciate the reviewer for the useful suggestion. We accepted the reviewer comment and removed from this revised manuscript the correlation analysis between the daily gross energy intake and estimated methane emission per cow result including the former figure 2 and its related information.

However, we kept the regression analysis (relationships among plasma urea concentration, milk urea concentration, and ruminal ammonia N as well as among methane emission and milk yield). Regarding the limited data, we respected the reviewer comment but we used a well-established a Latin square design (LSD) and in our experiment eight dairy cows in a replicated 4x4 LSD which has enough replications for the required data analysis (Nichols et al., 2019) for this study parameters.

  1. The high N retention values indicate either increase in body weight, or incorrect protocol to determine N intake- secretion- excretion. Please indicate in the table or in the body weights of cows (total or change) and comment on this if there was a problem in the protocol.

Response: We thank the reviewer for the valuable comment. Thus, our results support the reviewer’s assertion that the high N retention values indicate an increase in body weight. To clarify this, we included the body weight change data specifically the average daily gain (ADG) in Table 2. In addition, the following text “Cows fed with BhH had increased ADG (P = 0.02) followed by NGH and TTS as compared to NPH” is included in lines 310-311 of the revised manuscript. Moreover the following text was added in lines 326-328 of the revised manuscript, “Nitrogen retention showed a similar trend and was greater (P < 0.01) by 56.7%, 54.7%, and 67.6% for cows fed TTS, NGH, and BhH, respectively, than those fed NPH diet”.

Specific comments

  1. L30: introducing

Response: We thank the reviewer for the useful comment. We replaced “introduce” with “introducing” in line 31 of the revised manuscript. 

  1. L32: reduce

Response: We thank the reviewer for the useful suggestion. We believe using “reduce” instead of “reducing” is more consistent in line 33 of the revised manuscript.   

  1. L40 reduced

Response: We thank the reviewer for the useful comment. We replaced “reduce” with “reduction of” in line 40 of the revised manuscript. 

  1. L42 improving

Response: We thank the reviewer for the comment. We replaced “improve” with “improving” in line 42 of the revised manuscript. 

  1. L54 and throughout the manuscript P values should have a consistent form with P in capital letters italic, and the value in 2 digits after the point.

Response: We thank the reviewer for the useful comment. We used italicized capital P with values in 2 digits after the point throughout the revised manuscript.

  1. L61 protection against

Response: We thank the reviewer for the comment. We replaced “protection” with “protection against” in line 61 of the revised manuscript. 

  1. L66 enteric methane emission

Response: We thank the reviewer for the comment. We have added “enteric methane emission” at the end of the line 66 of the revised manuscript.

  1. L78: add “per unit of animal product” at the end of the line

Response: We thank the reviewer for the comment. We have added “per unit of animal product” at the end of the line 79 of the revised manuscript.

  1. L83 replace avoid with mitigate

Response: We thank the reviewer for the comment. We replaced “avoid” with “mitigate” in line 83 of the revised manuscript.

  1. L84 remove “is still”

Response: We thank the reviewer for the comment. We removed “is still” in the revised manuscript.

  1. L106 suggest rewording because it implies that you will be a comparing between treated and untreated teff straw

Response: We thank the reviewer for the comment. To avoid confusion, we rephrased the sentences“the effect of treating teff straw with effective microbes, molasses, and urea on the enhancement of the nutrient utilization of this available poor-quality roughage feed and performance of lactating dairy cows” given in lines 109-111 of the revised manuscript.

  1. L111 reduce methane emission compared with…

Response: We thank the reviewer for the comment. We added “natural pasture hay” after “reduce methane emission compared with” in lines 113-114 of the revised manuscript.

  1. L127 Please provide days in milk at the start of the project

Response: We thank the reviewer for the comment. We provided days in milk of the experimental dairy cows “120 ± 18 days in milk” in lines 129 of the revised manuscript.

  1. L145 dried to replace wilted

Response: We thank the reviewer for the comment. We replaced “wilted” with “air-dried” in line 151 of the revised manuscript.

  1. L146 mixed

Response: We thank the reviewer for the comment. We added “mixed” in the following sentence “The harvested forage was air dried for 2-3 days in the field and mixed before storage for the feeding trial” in lines 151-152 of the revised manuscript.

  1. L154-160 On average, at 240 kg body weight, DMI should be calculated at 7.2 kg/d. Were the diet formulates so that such a level of intake would meet the nutrient requirement to produce 4 kg milk/d? Please put in details

Response: We appreciate the reviewer for this useful comment. Initially, we assumed 3% of the body weight intake. Based on this, we revised as “To achieve the targeted milk yield (4 kg milk/d), the TMR diets were calculated based on NRC [1] for the maintenance and lactation requirement of the lactating dairy cow (average body weight, 238.3 kg) for each treatment group” given in lines 167-170 of the revised manuscript.

  1. Table 1 Identify abbreviations

Response: We thank the reviewer for the comment. We used the full name of the abbreviation “TMR” as “total mixed ration” in line 176 of the revised manuscript.  

  1. Table 1 Is this on DM basis? please specify. Also report ADL on NDF basis. Also add correct units of NEm and NEL and reference of calculations

Response: We thank the reviewer for the useful comment. Yes, we used on DM basis and it was specified as “g/kg of DM” for the parameters presented in Table 1 such as DM, OM, CP …etc., given in lines 177-178 in Table 1 footnote. We also reported “The ADL/NDF ratio of the dietary treatments for NPH, TTS, NGH and BhH is 0.13, 0.17, 0.22 and 0.25, respectively”, given in lines 173-174. The units of NEm and NEL was corrected as Mcal/kg DM by converting from MJ/kg, given in lines 179-180 in Table 1 footnote and the reference of calculations was cited in Equations (2) & (3) in lines 226-227 of the revised manuscript.

  1. L181 replace by “treatment group” with “cow”

Response: We thank the reviewer for the comment. We replaced “treatment group” with “cow” in line 188 of the revised manuscript.

  1. L184, L189, L192 what time the sampling was done? was it spread throughout the day or at specific time each day?

Response: We appreciate the reviewer for the useful comment. We added specific time of the sample collection. Thus, for milk sample “During each experimental period (from 15 to 21 days), milk samples were collected at each milking time (08:00 h and 16:00 h)” in line 188. For urine sample, “three times per day (07:00 h, 13:00 h and 17:00)” in Line 192. For fecal sample “in the morning (07:00 h), afternoon (13:00 h) and evening (17:00 h)” in line 197 of the revised manuscript.

  1. L256 there is no details on chromium oxide use in the previous text. Please add

Response: We thank the reviewer for the valuable comment. We added the detail information that “Chromium oxide was given daily (5 g for each cow) in the morning feeding on days 15 through 21 of each experimental period” in lines 263-264 of the revised manuscript.

  1. L301-302: Please separate results of intake from results of digestibility. It is rather confusing in this form

Response: We appreciate the reviewer for the useful comment. We separated the results of intake and digestibility as “Cows fed with BhH had the highest (P < 0.05) DM, OM, and CP intake followed by NGH and TTS” in lines 309-310 and “Likewise, cows fed with BhH, NGH and TTS had higher (P < 0.05) DM, OM, and CP diet digestibility than NGH” in lines 309-310 of the revised manuscript.

  1. L301-302 here it is CP digestibility while it is N digestibility in the table. Please correct.

Response: We thank the reviewer for the comment. We replaced “CP” with “N” in line 311 of the revised manuscript.

  1. L302 tended? I see it is significant please correct and add P value

Response: We thank the reviewer for the comment. We added “P value (P<0.01)” in line 310 of the revised manuscript.

  1. L307 here and throughout the manuscript: in table title, full name of treatment should

be spelled out

Response: We appreciate the reviewer for the useful comment. We accepted and applied the full name of treatment “natural pasture hay, treated teff straw, napier grass hay and brachiaria hybrid grass hay” in all tables title throughout the revised manuscript.

  1. Table 3: what is the difference between milk N/ N intake and MNE? Please define MNE in the table footnote

Response: We appreciate the reviewer for the useful comment. The milk nitrogen efficiency (MNE) = (N milk / N intake) × 100% to express in percentage while milk N/ N intake is in ratio. So, we removed the MNE and kept milk N/ N intake to be consistence with other parameters reported in ratio as shown in Table 3 of the revised manuscript. 

  1. Table 4: Ruminal pH and plasma concentrations…. Rumen pH values should be in 2 decimals

Response: We thank the reviewer for the comment. We presented the values for the parameters (ruminal pH, plasma concentrations…etc.) in in Table 4 in 2 decimals after the point in the revised manuscript.

  1. L347 remove that

Response: We thank the reviewer for the comment. We removed “that” as suggested from line 356 in the revised manuscript.

  1. L352 higher milk urea nitrogen

Response: We thank the reviewer for the comment. We added “urea” as suggested to complete the phrase as “higher milk urea nitrogen” in line 362 in the revised manuscript.

  1. L368 previous comment regarding correlation data. GEI is similar DMI, and you are using a formula to calculate CH4 based on DMI or GEI, for sure there will be correlation previous comment regarding correlation data. GEI is similar DMI, and you are using a formula to calculate CH4 based on DMI or GEI, for sure there will be correlation

Response: We appreciate the reviewer for the useful comment. The correlation analysis result for GEI and DMI was removed as described in the same comment from the reviewer in no. 4 above.

  1. L393 avoid abbreviation at the beginning of the sentence

Response: We thank the reviewer for the comment. We used the full name of the abbreviation “DMI” as “Dry matter intake” at the beginning of the sentence in line 395 of the revised manuscript.

  1. L397 remove potential

Response: We thank the reviewer for the comment. We removed “potential” as suggested from line 399 in the revised manuscript.

  1. L400-402 hard to understand. Please separate into 2 or 3 sentences o reword accordingly.

Response: We appreciate the reviewer for the useful comment. We modified as “These differences in fiber digestibility of TMR diets might be related to the quality of the dietary NDF ingested (Christensen et al., 2015)” in lines 402-403 of the revised manuscript.

  1. L405-406 urea and molasses added to the tef straw silage

Response: We thank the reviewer for the comment. We added “and” in between urea and molasses as suggested to complete the phrase as “urea and molasses added to the tef straw silage” in line 405 in the revised manuscript.

  1. L408-410 also there is a positive relationship between DMI and fecal N excretion. Please use this to enrich discussion

Response: We thank the reviewer for the comment. However, it seemed to us that the positive relationship between DMI and fecal N excretion was not supported in this study in which the cow feed NPH diet (with low DMI) had higher fecal N excretion as compared to other TMR diets (Table 3) in the revised manuscript.

  1. L420-421 This is not clear, please reword and be more specific

Response: We appreciate the reviewer for the useful suggestion. It was modified as “This indicates that inclusion of improved forages in the TMR diets of dairy cows improve nitrogen efficiency” in lines 424-425 in the revised manuscript.

  1. L426 rumen NH3 concentration rather rumen fermentation characteristics

Response: We thank the reviewer for the comment. We replaced “rumen fermentation characteristics” with “rumen NH3 concentration” in line 426 of the revised manuscript.

  1. L428-431 but this is against the absence of diet effect on rumen NH3 concertation. Please elaborate on this

Response: “We attribute the lower plasma urea nitrogen concentrations recorded for the BhH and NGH diets to better utilization of nitrogen in the rumen, because the end-product of protein in the rumen is ammonia N, then ammonia N is absorbed by rumen epithelial cells and transferred to liver to form urea, which is released to blood with the form of blood urea (Getahun et al., 2019). This might be due to the higher values of ruminal NH3-N in BhH and NGH diets were reflecting the higher plasma urea N in this study” in lines 428-433 of the revised manuscript.

  1. L435-436 This is a speculation not measured. Please specify

Response: We thank the reviewer for the comment. We removed the ideas that lead to speculation “which in turn increases the availability of nutrients for ruminal fermentation and results in higher concentrations of propionic acid, which can be converted to glucose in the liver through gluconeogenesis” in the revised manuscript.

  1. L440-443 add reference

Response: We thank the reviewer for the comment. Accordingly, we added the reference (Jorjong et al., 2014) in line 443 in the revised manuscript.

  1. L443-444 how is this related to your findings, please elaborate.

Response: We appreciate the reviewer for the useful suggestion. We removed it “other authors have reported high concentrations of NEFA and BHBA during mid-lactation in dairy cows” from the revised manuscript.

  1. L448 plasma urea concertation

Response: We thank the reviewer for the comment. We added “urea” in between plasma and concertation as suggested to complete the phrase as “plasma urea concertation” in line 470 in the revised manuscript.

  1. L452-455 mainly due to high forage to concentrate ratio (70:30).

Response: We appreciate the reviewer for the useful comment. We incorporated it in the discussion part supported with reference as “This might be mainly due to high forage to concentrate ratio (70:30)” in lines 455-456 in the revised manuscript.

  1. L463-464 This is irrelevant. Please consider removing or rewording

Response: We appreciate the reviewer for the useful suggestion. We removed it “This variation might be due to the TMR feeding system used in this study that can increase the precision-feeding of dairy cows” from the revised manuscript. 

  1. L468-470 it is HIGHLY unlikely that 1 kg of molasses used in 50kg of hay to exert such an effect. Most probably it is due to a positive energy status of the cow due to higher DMI. Please re-investigate

Response: We are grateful to thank the reviewer for this comment and we agreed. It was revised as “The highest plasma glucose concentration recorded for BhH TMR diet in this study might be due to a positive energy status of the cow related to higher DMI (Drackley and Cardoso, 2014)” in lines 435-437 in the revised manuscript.

  1. 50. L477-479 unclear, please reword

Response: We appreciate the reviewer for the useful comment. We revised as “Thus, the methane production variation in our study among the dietary treatments might be related to the difference in the proportion of nutrient such as the high NDF content of the NPH-based TMR” in lines 474-476 in the revised manuscript.

  1. L48-suggest replacing over with “which contribute to”

Response: We thank the reviewer for the comment. We replaced “over” with “which contribute to” in line 478 of the revised manuscript.

  1. L491 Suggest adding: The environmental cost of producing NGH, bhH forages should be examined before recommending using this strategy to mitigate CH4 emissions and urinary N excretion.

Response: We are grateful to thank the reviewer for this suggestion. We added to enrich our discussion as “Moreover, the environmental cost of producing NGH and BhH forages should be examined before recommending using this strategy to mitigate CH4 emissions and urinary N excretion” in lines 483-485 of the revised manuscript.

  1. In the conclusion, please use full name of treatment rather than the abbreviation, at least when first mentioned.

Response: We thank the reviewer for the important comments. We used the full name of the diet treatment abbreviation asDairy cows fed treated teff straw (TTS), natural grass hay (NGH) and brachiaria hybrid grass hay (BhH) had better feed intake and nutrient utilization as well as milk yields than cows fed natural pasture hay (NPH)” in lines 492-494 of the revised manuscript.

  1. In the conclusion: The key point in the conclusion that increasing production reduced enteric CH4 emission, which should be the focal point of reducing GHG emissions in countries like Ethiopia where nutrition is limitation

Response: We are grateful for this valuable comment as it points to an important rationale of this study and we tried to focus and capitalize on it in our conclusion. Consequently, we added in the last paragraph of the conclusion as “Consequently, this study highlighted the possibility of increasing milk production and reducing greenhouse gas emissions in countries like Ethiopia where nutrition is limitation” in lines 503-504 of the revised manuscript.

  1. L494-495 This is repeated in the next phrase, please remove one of the repeated phrases or reword

Response: We appreciate the reviewer for the useful suggestion. Accordingly, we modified by merging the two sentences as “Feeding TTS, NGH and BhH hay as a basal diet changed the N excretion pathway from urine to feces, indirectly mitigated ammonia emissions, thus potentially benefitting the environment” in lines 494-496 of the revised manuscript.

References

Christensen, R., Yang, S., Eun, J.-S., Young, A., Hall, J. & MacAdam, J. 2015. Effects of feeding birdsfoot trefoil hay on neutral detergent fiber digestion, nitrogen utilization efficiency, and lactational performance by dairy cows. Journal of dairy science, 98, 7982-7992.

Drackley, J. K. & Cardoso, F. 2014. Prepartum and postpartum nutritional management to optimize fertility in high-yielding dairy cows in confined TMR systems. Animal, 8, 5-14.

Getahun, D., Getabalew, M., Zewdie, D., Alemneh, T. & Akeberegn, D. 2019. Urea Metabolism and Recycling in Ruminants. Biomedical Journal of Scientific & Technical Research, 20, 14790-14796.

Jorjong, S., Van Knegsel, A., Verwaeren, J., Lahoz, M. V., Bruckmaier, R., De Baets, B., Kemp, B. & Fievez, V. 2014. Milk fatty acids as possible biomarkers to early diagnose elevated concentrations of blood plasma nonesterified fatty acids in dairy cows. Journal of dairy science, 97, 7054-7064.

Nichols, K., Dijkstra, J., Van Laar, H., Pacheco, S., Van Valenberg, H. & Bannink, A. 2019. Energy and nitrogen partitioning in dairy cows at low or high metabolizable protein levels is affected differently by postrumen glucogenic and lipogenic substrates. Journal of dairy science, 102, 395-412.

Reviewer 2 Report

Interesting study.

Grammatical errors and formatting need to be fixed throughout the manuscript.

Simple summary

Suggest re-phrasing of sections to improve clarity e.g. Line 29-30; ‘Feeding poor quality roughage for dairy cows is common practice in tropical regions that decrease productivity and increase methane emissions.’ could be changed to ‘In tropical regions it is common practice to feed dairy cows poor quality roughage but this diet has been shown to decrease animal productivity and increase methane emissions.’

Abstract

Line 62-63: Re-phrase to clarify that methane emissions are estimations rather than measurements e.g. ‘Estimated methane yields per dry matter intake, dietary energy intake and milk yield were decreased in dairy cows fed BhH, NGH and TTS diets when compared to cows fed a NPH diet (P<0.05).’

Introduction

Line 107: Define the term ‘effective microbes’. Could also be useful to add a line describing why treatment of teff straw with microbes, molasses and urea would be expected to improve nutrient utilization etc.

Material & methods

Need to include information on ethical approval for the animal study.

Line 127-128: Clarify if the 8 cows were in 8 separate pens or the 8 cows were placed in one pen to separate them from the rest of the herd.

Line 130-134: The description of the study design is a little confusing. N=2 for each diet for each time frame? This is a low number of reps per treatment. A diagram may be helpful.  

Line 149: Define the term 'effective microbes'.

The abbreviation format in Table 1 is confusing, could it be given in order of appearance rather than alphabetical order? Recommend placing a line between ‘Feed ingredients’ and ‘Treatment diets’ to improve readability of the table.

Line 201-203: Body weight measured at the start, middle and end of each 21 day block or at the start, middle and end of the 12 weeks? 

Results

Abbreviation heavy- would be easier to follow if whole words were used again before being abbreviated.

Tables- Again could abbreviations in footnote appear in alphabetical order?  

Table 4 could be moved to supplementary?

Figure 1b is mentioned in text before figure 1a.

Line 362- 373: It should be made clear that these are estimated methane yields calculated using a formula based on DMI, not real measurements.

Table 6 and figure 2 could be moved to supplementary?

Discussion/Conclusion

Need clear caveats regarding the low number of reps per diet and (in the conclusion) the use of methane yield estimation rather than measurements.

Author Response

Reply to Reviewer #2 comments

  1. Grammatical errors and formatting need to be fixed throughout the manuscript.

Response: We thank the reviewer for the valuable comment. The grammatical errors and formatting corrected throughout the manuscript. To this end the input from senior co-authors was significant.

Simple summary

  1. Suggest re-phrasing of sections to improve clarity e.g. Line 29-30; ‘Feeding poor quality roughage for dairy cows is common practice in tropical regions that decrease productivity and increase methane emissions.’ could be changed to ‘In tropical regions it is a common practice to feed dairy cows poor quality roughage but this diet has been shown to decrease animal productivity and increase methane emissions.’

Response: We appreciate the reviewer for the useful suggestion. We accepted the suggestion as “In tropical regions, it is a common practice to feed dairy cows poor quality roughage but this diet has been shown to decrease animal productivity and increase methane emissions” in lines 29-31 in the revised manuscript

Abstract

  1. Line 62-63: Re-phrase to clarify that methane emissions are estimations rather than measurements e.g. ‘Estimated methane yields per dry matter intake, dietary energy intake, and milk yield were decreased in dairy cows fed BhH, NGH and TTS diets when compared to cows, fed a NPH diet (P<0.05).’

Response: We appreciate the reviewer for the useful suggestion. We accepted the re-phrase and included in the revised manuscript in lines 62-64 as “Estimated methane yields per dry matter intake, dietary energy intake and milk yield were decreased in dairy cows fed BhH, NGH and TTS diets, when compared to cows, fed a NPH diet (P<0.05)”.

Introduction

  1. Line 107: Define the term ‘effective microbes’. Could also be useful to add a line describing why treatment of teff straw with microbes, molasses and urea would be expected to improve nutrient utilization etc.

Response: We thank the reviewer for the valuable comment. We added the definition of ‘effective microbes’ and their importance as Thus, treating these feed resource using bio-chemical such as effective microbes (a liquid mixture of important beneficial microorganisms) and urea molasses improved feed digestibility and nutritive value of straw (Alemu et al., 2020)” in lines 106-108 of the revised manuscript.

Material & methods

  1. Need to include information on ethical approval for the animal study.

Response: We appreciate the reviewer for the useful suggestion. “All animal care and handling procedures were reviewed and the experimental protocol was approved by Andassa Livestock Research center prior to conducting the experiment and the animals were under constant observation of veterinarians” in lines 137-140 of the revised manuscript.

  1. Line 127-128: Clarify if the 8 cows were in 8 separate pens or the 8 cows were placed in one pen to separate them from the rest of the herd.

Response: We thank the reviewer for the comment.The lactating dairy cows were placed in an individual separate pen” in lines 131 of the revised manuscript.

  1. Line 130-134: The description of the study design is a little confusing. N=2 for each diet for each time frame? This is a low number of reps per treatment. A diagram may be helpful.  

Response: We understand the concern of the reviewer. However, a Latin square design is a well-established design (Nichols et al., 2019) in the arrangement of 4 treatments, each one repeated 4 times with two cows. Thus, in this study, we used “a replicated 4x4 Latin square which gives a total of 32 experimental units for treatment” in lines 133-135 of the revised manuscript.  

  1. Line 149: Define the term 'effective microbes'.

Response: We thank the reviewer for the comment. We already defined it. Please see in the above response in number 5.

  1. The abbreviation format in Table 1 is confusing; could it be given in order of appearance rather than alphabetical order? Recommend placing a line between ‘Feed ingredients’ and ‘Treatment diets’ to improve readability of the table.

Response: We appreciate the reviewer for the suggestion. We put the abbreviation in a footnote in order of appearance for all tables thought out the revised manuscript. In addition, we placed a line between ‘‘Feed ingredients’ and ‘Treatment diets’’ in table 1 of the revised manuscript.

  1. Line 201-203: Body weight measured at the start, middle and end of each 21 day block or at the start, middle and end of the 12 weeks? 

Response: We thank the reviewer for the useful comment. We revised thatThe body weight (BW) of each dairy cow was weighed at the start, middle, and end of each period for the whole experiment time” in lines 208-209 of the revised manuscript.

Results

  1. Abbreviation heavy- would be easier to follow if whole words were used again before being abbreviated.

Response: We thank the reviewer for the valuable comment. We used the full name of the abbreviation “TMR” as “total mixed ration” in line 306 of the revised manuscript.

  1. Tables- Again could abbreviations in footnote appear in alphabetical order?  

Response: We appreciate the reviewer for this suggestion again. We put the abbreviation in a footnote in order of appearance as described in the above response number 10.

  1. Table 4 could be moved to supplementary?

Response: We thank the reviewer for the comment. Yet, the result presented in Table 4 referred many times in the text. So, we prefer to keep it in the main document.

  1. Figure 1b is mentioned in text before figure 1a.

Response: We appreciate the reviewer for the suggestion. We adjusted the alignment of the figure and figure 1a (in line 344) was mentioned before 1b (inline 363) in the revised manuscript.

  1. Line 362- 373: It should be made clear that these are estimated methane yields calculated using a formula based on DMI, not real measurements.

Response: We appreciate the reviewer for the useful suggestion. We used the word “estimated” before “methane emission” through the lines from 372-381 of the revised manuscript.  

  1. Table 6 and figure 2 could be moved to supplementary?

Response: We thank the reviewer for the comment. We already removed figure 2 based on the other reviewer comment. But for Table 6 since the results presented in it referred many times in the text, we prefer to keep in the main document.

Discussion/Conclusion

  1. Need clear caveats regarding the low number of reps per diet and (in the conclusion) the use of methane yield estimation rather than measurements?

Response: Again, we understand the reviewer's concerns. Regarding the number of replication per diet since our major objective is to assess the effect of diet on milk yield, nitrogen utilization, and estimated methane emission, eight dairy cows in replicated LSD has many replications enough for the required analysis (Nichols et al., 2019) as described in the above in response to comment number 8. For the second concern, we emphasize the estimated methane emission rather than measured methane yield than measurement methane throughout the revised manuscript.

References

Alemu, D., Tegegne, F. & Mekuriaw, Y. 2020. Comparative evaluation of effective microbe–and urea molasses–treated finger millet (Eleusine coracana) straw on nutritive values and growth performance of Washera sheep in northwestern Ethiopia. Tropical Animal Health and Production, 52, 123-129.

Nichols, K., Dijkstra, J., Van Laar, H., Pacheco, S., Van Valenberg, H. & Bannink, A. 2019. Energy and nitrogen partitioning in dairy cows at low or high metabolizable protein levels is affected differently by postrumen glucogenic and lipogenic substrates. Journal of dairy science, 102, 395-412.
